# Prediction of the Delamination at the Steel and CFRP Interface of Hybrid Composite Part

**DOI:** 10.3390/ma14216285

**Published:** 2021-10-21

**Authors:** Jun-Su Park, Jae-Hong Kim, Joon-Hong Park, Dae-Cheol Ko

**Affiliations:** 1Department of Nanomechatronics Engineering, Pusan National University, Busan 46241, Korea; wnstn791@pusan.ac.kr; 2ERC for Innovative Technology on Advanced Forming, Pusan National University, Busan 46241, Korea; kjh86@pusan.ac.kr; 3Department of Mechanical Engineering, Dong-A University, Busan 49315, Korea; acttom@dau.ac.kr

**Keywords:** carbon-fiber-reinforced plastic (CFRP), steel/CFRP hybrid part, prepreg compression molding (PCM), cohesive zone model (CZM), delamination

## Abstract

The purpose of this study was to predict the adhesive behavior of steel and carbon-fiber-reinforced plastic (CFRP) hybrid parts based on the cohesive zone model (CZM). In this study, the steel sheet and CFRP were joined by epoxy resin in the CFRP prepreg during the curing process, which could generate delamination at their interface because of the springback of steel or the thermal contraction of the CFRP. First, double cantilever beam (DCB) and end-notched flexure (ENF) tests were performed to obtain various adhesion properties such as the critical energy release rate of mode I, mode II (*G_I_*, *G_II_*), and critical stress (*σ_max_*). A finite element (FE) simulation was performed to predict delamination using CZM, which was also used to describe the interfacial behavior between the steel sheet and the CFRP. Finally, a U-shape drawing test was performed for the steel/CFRP hybrid parts, and these results were compared with analytical results.

## 1. Introduction

Recently, regulations on fuel efficiency and emissions have been tightened in the automotive industry owing to environmental pollution and the depletion of fossil fuels. As a result, the automotive industry has been focusing on manufacturing lightweight parts using magnesium, composite materials, advanced high-strength steel (AHSS), and carbon fiber reinforcement plastic (CFRP). CFRP has been widely used because of its superior strength, stiffness, and fatigue performance compared with conventional lightweight metals. However, the application of CFRP could lead to high production costs and low productivity [1,2,3].

Recent studies have focused on the manufacturing of steel/CFRP hybrid parts to solve the aforementioned problems. Wang et al. manufactured a hybrid part by forming a prepreg on an existing steel sheet using resin transfer molding (RTM) [4]. Kim et al. manufactured a hybrid part using RTM and applied adhesive bonding between the steel part and the CFRP [5]. Kim et al. manufactured a hybrid B-pillar outer panel with steel and CFRP reinforcements using vacuum-assisted resin transfer molding [6]. However, these studies require an additional joining process to assemble existing metal components. Therefore, prepreg compression molding (PCM) for the hybrid part was applied to improve productivity and product quality [7].

In the PCM process, fractures at the interface between dissimilar materials can occur owing to different material behaviors and mechanical properties. Therefore, it is important to predict the adhesive behavior at the interface between the steel and the CFRP. Quagliato et al. evaluated the bonding strength of dissimilar materials in the forming process of sandwich panels consisting of steel and CFRP [8]. Kim et al. proposed a technique to measure the energy release rate at the adhesive interface of dissimilar materials and investigated the causes of changes in the energy release rate [9]. Lee et al. predicted the delamination behavior between the steel and the CFRP based on the cohesive zone model (CZM) [10]. Zhao et al. predicted the delamination phenomena of composites under out-of-plane loading using the extended finite element method (XFEM) [11]. Funari et al. proposed a computational formulation based on a moving mesh methodology and interface modeling to predict the delamination in multilayered composite beams [12]. Huang et al. described a novel method for analyzing inter-laminar delamination for a laminate of which advantages were a minimum number of input data and no iteration [13]. However, most of these studies did not deal with delamination during the forming process of the hybrid part.

The purpose of this study was to predict the adhesive behavior in the forming process for steel/CFRP hybrid parts based on the finite element (FE) simulation with the CZM. Here, the steel sheet and CFRP were joined by epoxy resin in the CFRP prepreg during the curing process. During the forming process of the hybrid parts, delamination can occur owing to the springback of steel, lack of adhesion, and thermal contraction of the CFRP. First, double cantilever beam (DCB) and end-notched flexure (ENF) tests were performed to obtain various adhesion properties, such as the critical energy release rate and critical stress. FE simulation was performed to predict the adhesive behavior with the CZM, which was used to describe the interfacial behavior between the steel and the CFRP. Finally, FE simulations and experiments for the U-shape drawing of the hybrid prat were performed to predict the delamination at the steel and CFRP interface in the forming process for verification.

## 2. Evaluation of the Bonding Characteristics for the Steel/CFRP Part

The delamination at the interface is primarily caused by the tensile direction (mode I) and shear direction (mode II). Additionally, the energy release rate of each mode must be derived to predict the actual delamination because of the mixed direction [14,15]. In this study, double cantilever beam (DCB) and end-notched flexure (ENF) tests were carried out to evaluate the bonding characteristics of the steel/CFRP part.

Specimens were manufactured using a galvannealed DP980 sheet (POSCO, Pohang, South Korea) and CFRP (SK Chemicals, Seongnam, South Korea) with a thickness of 1.2 mm each, where the CFRP consisted of six plies of twill weave prepreg. The CFRP specimens were fabricated in the 0° fiber direction. The specimens were cured for 3 min at 160 °C. A release film of which thickness was 0.02 mm was inserted between the steel and CFRP to ensure a constant initial crack length for each test.

### 2.1. Double Cantilever Beam (DCB) Test

The DCB test has been widely used to evaluate the fracture toughness of composite laminates for mode I. The energy release rate (*G_I_*) for mode I can be derived based on Griffith’s linear elastic fracture mechanics by measuring the load, displacement, and delamination length. Experiments were conducted according to the ASTM 5528-13 standard [16], and the dimensions of the specimens are shown in Figure 1. A release film of 70 mm was used to obtain the initial crack length. A ruler with 1 mm intervals was attached to the side of the specimen to measure the delamination length.

Figure 2 illustrates a griped specimen on the INSTRON universal testing machine (INSTRON 5566A, Instron, Norwood, MA, USA) during the test. The test was conducted with a crosshead velocity of 5 mm/min, and the delamination length was measured on one side of the specimen using a camera. Figure 3 shows the derived load–displacement curves and failure surface for specimen of DCB test.

*G_I_* is generally calculated using the modified beam theory (MBT), compliance calibration method (CCM), and the modified compliance calibration method (MCCM). According to the MBT theory, the energy release rate, *G_I_*, is as follows:(1)$$G_{I}=\frac{3P_{t}\delta_{t}}{2B\;\left(a_{0}+\left|∆\right|\right)}$$
where $P_{t}$, $\delta_{t}$, B, and $a_{0}$ represent the maximum load, the deflection corresponding to the load, the width of the specimen, and the length of the interlayer separation, respectively. ∆ is experimentally determined by plotting the least-squares of the cube root of compliance (*C*^1/3^) with respect to the delamination length.

According to the CCM theory, $G_{I}$ is as follows:(2)$$G_{I}=\frac{nP_{t}\delta_{t}}{2Ba_{0}}$$
where n is the slope of lnC and lna.

According to the MCCM theory $G_{I}$ is as follows:(3)$$G_{I}=\frac{3P_{t}{}^{2}C^{2/3}}{2A_{1}Bh}$$
where *h* is the thickness of the specimen, and *A_1_* is the slope of *C^1/3^* and *a_0_/h*.

### 2.2. End-Notched Flexure Test

The ENF test was performed to derive the energy release rate (*G_II_*) for mode II. The test was conducted according to the ASTM D7905/D7905M-19 standard [17], and the dimensions of the specimens are shown in Figure 4. A release film was used to obtain an initial crack length of 10 mm and a ruler with 1 mm intervals was attached to the side of the specimen to measure the delamination length, similar to the DCB test. The experiment was performed at a punch velocity of 2 mm/min. Figure 5 illustrates the experimental setup with the MTS universal testing machine (MTS Landmark^TM^ 100 kN, MTS Systems Corporation, Eden Prairie, MN, USA). Figure 6 shows the derived load-displacement curves and failure surface for specimen of ENF test.

The flexural modulus is expressed by Equation (4), and the energy release rate of mode II is expressed by Equations (5) and (6).
(4)$$E_{f}=\frac{2L^{3}+3a_{0}{}^{3}}{8Bh_{1}{}^{3}C}$$
(5)$$G_{II}=\frac{9a_{0}{}^{2}P_{C}{}^{2}}{16B^{2}E_{f}h_{1}{}^{3}}$$
(6)$$\;G_{II}=\frac{9a_{0}{}^{2}P_{C}\delta_{C}}{2B\left(2L^{3}+3a_{0}{}^{3}\right)}$$
where *B* is the specimen width, $h_{1}$ is half of the specimen thickness, L is the span length, $a_{0}$ crack length of the specimen, $P_{C}$ is the load at the crack occurrence, and $\delta_{C}$ is the bending displacement at the crack occurrence.

### 2.3. Experimental Results

Figure 7 shows the derived R-curve based on the results of DCB and ENF tests. *G_I_* and *G_II_* for steel/CFRP are also summarized in Table 1. The deviation of experimental results, as shown in Figure 3a and Figure 6a, was caused by the difference in surface conditions for steel sheets and the unstable adhesive layer at crack initiation. As a result, *G_I_* was about 93% lower than *G_II_*. The fracture of the DCB test only occurred at the adhesive layer, which was resin in the prepreg, as shown in Figure 3b. On the other hand, the fracture of the ENF test appeared to have occurred in both the adhesive layer and the coating layer of the DP980 sheet, as shown in Figure 6b. This means that the adhesion of steel/CFRP by resin is more vulnerable to tensile direction failure than shear direction failure.

## 3. Cohesive Parameter for Steel/CFRP Hybrid Part

### 3.1. Cohesive Zone Model(CZM)

In this study, the interfacial behavior of the steel/CFRP hybrid part was described via FE simulation using the CZM. This method has been extensively used for several years mainly to simulate the delamination of composite materials and the cracking progression in adhesion joints [18,19]. However, the limitation of using the CZM is that it is difficult to accurately determine the required input parameters; therefore, repetitive simulations are required based on the experimental results.

The FE simulation with CZM was conducted using the ABAQUS (Version 2020, Dassault Systèmes, Vélizy-Villacoublay, France), and the traction-separation law, as shown in Figure 8, was applied for the analysis. Once the stress reaches the point of cohesive stress, called damage initiation, material damage occurs, and its damaging behavior continues until the material is filed by the damage evolution law [20]. The total area under the traction-separation graph represents the energy required for adhesion separation, which is known as the energy release rate. In the FE simulation, the directional energy release rates are essential input parameters. In addition, the slope (penalty stiffness, *K*) until point $t_{n}^{0}$ or $t_{t}^{0}$ and the peak stress are required to accurately define the traction-separation law. The penalty stiffness is expressed by Equation (7).
(7)K=E/t
where E and *t* are the elastic modulus of the resin and thickness of the adhesive, which is the same as the thickness of release film (0.02 mm).

In this study, the adhesive layer was assumed to be isotropic and peak stress ($t_{n}^{0}$ or $t_{t}^{0}$) was defined as the same with the tensile and shear strength of resin. The material properties of the CFRP, resin, and steel used in the analysis are summarized in Table 2.

### 3.2. Conditions for FE Simulation with CZM

In the FE model for the DCB and ENF test, the steel and CFRP were modeled using the four-node bilinear plane strain element (CPE4R), and adhesive was modeled using the four-node two-dimensional cohesive element (COH2D4) for the efficiency of the simulation. Additionally, the mesh size of steel/CFRP was adjusted to be square to improve the convergence of the analysis. Isotropic elastic behavior was considered for steel and CFRP, and traction behavior was applied for the bonding interface.

The load in the DCB test was applied for the vertical direction of the nodes located at the upper hinges. The lower hinge was fixed in vertical and horizontal directions and rotated freely along the horizontal axis. The conditions for the modeled DCB test are shown in Figure 9a.

The experimental devices were modeled for the numerical application of the ENF test. The shells for two spans were modeled for supporting the specimen and the shell, and one pin was modeled for applying the load. Between the two spans and the specimen, the contact condition was the surface-to-surface (standard) contact option. One pin and specimens were tied to prevent slip or separation. The FE model for the ENF test is shown in Figure 9b.

### 3.3. Results of the FE Simulation

Figure 10 shows the results of the FE simulation for the DCB test, compared with the experimental results. The MBT, CCM, and MCCM are not significantly different in the load-displacement graph because of the similar peak stress for each theory. Therefore, a conservative value, which was calculated by the CCM, was applied to the FE simulation of the U-shape drawing for the steel/CFRP hybrid part. The analytical results showed good agreement with the experimental data; therefore, the interfacial behavior for the tensile load could be predicted well.

Figure 11 shows a comparison of the results of the FE simulation for the ENF test with the experimental results. Figure 10b shows that the similarity in the results of the FE simulation and the experiment, and the linear behavior of the specimen up to the applied displacement of 4.3 mm. The experimental results showed a lower load than the analysis results, in which the inconsistency was attributed to the invariability of the adhesive thickness in the FE simulation. Interfacial behavior in the shear load could still be predicted within a maximum error of 9.0%.

## 4. U-Shape Drawing for Steel/CFRP Hybrid Part

### 4.1. Experiment for the U-Shape Drawing

Figure 12 shows a schematic diagram of the manufacturing process: (a) six plies of CFRP prepreg were stacked on a DP980 sheet of thickness 1.2 mm, (b) the release agent was spread on the tools to prevent adhesion between the tools and the CFRP, (c) the tools were heated up to 160 °C by a cartridge heater, (d) the steel blank with the CFRP prepreg patch was held by a blank holder, (e) the hybrid part was successively deformed and cured for 3 min, (e) and finally, the steel/CFRP hybrid part was ejected from the mold.

As shown in Figure 13, the steel/CFRP part was experimentally manufactured using a stamping toolset of a draw type, which consists of a punch, lower die, pad, and blank holder. A servo press with a capacity of 2000 kN was used in this experiment. The temperature required for the forming and curing processes was maintained using tools with heating channels.

Figure 14 shows the result of the experiment for the U-shape drawing. The endpoint of steel was moved by approximately 4.71 mm in the x direction and 8.14 mm in the y direction. Further, the endpoint of the CFRP was moved by approximately 3.23 mm in the x direction and 1.3 mm in the y direction. The delamination length was measured using the attached ruler on the CFRP and through the microscope with 10× magnification and was found to be approximately 20.84 mm, as shown in Figure 15.

### 4.2. FE Simulation for the U-Shape Drawing

The FE model for the U-shape drawing was designed to predict the interfacial behavior by the springback of steel, as shown in Figure 15. The element type and mechanical properties of steel, CFRP, and adhesive were applied to the same data as in Section 3.2. The bottom of the steel was fixed, and the displacement for the amount of springback in the experiment was input at the end of the steel to describe the material behavior.

Figure 16 shows the result of the FE simulation for the U-shape drawing. Delamination was initiated when the endpoint of the steel was moved by approximately 2.04 mm in the x direction and 2.58 mm in the y direction. In addition, a crack was propagated according to the increase in the deformation of the steel. Finally, the delamination length in the FE simulation was approximately 15.13 mm as against 20.84 mm in the experiment. This difference was caused by the deformation path of the steel and the thermal contraction of the CFRP; nevertheless, this study verifies the effectiveness to predict the adhesive behavior of the CZM for manufacturing steel/CFRP hybrid parts. The accuracy of prediction could be improved by considering the changeable thermomechanical properties for the curing behavior of the CFRP and through coupled analysis for forming and springback.

## 5. Conclusions

In this study, the adhesive behavior of the steel/CFRP hybrid part was predicted based on the FE simulation with the CZM. DCB and ENF tests were carried out to evaluate the bonding characteristics of the steel/CFRP part. GI and GII were calculated based on these tests, and the results indicated that the adhesion of the steel/CFRP with resin was more vulnerable to tensile direction failure than shear direction failure. The interfacial behavior of the steel/CFRP hybrid part was described by FE simulation using the CZM. Cohesive parameters for the CZM were determined by comparing the experimental and analytical results, which showed good agreement within a maximum error of 9.0%. The FE simulations and experiments for the U-shape drawing of the hybrid prat were performed to predict the delamination at the steel and CFRP interface in the forming process for verification. The delamination length was predicted to be approximately 15.13 mm in the FE simulation as against 20.84 mm in the experiment. Therefore, this study verifies the effectiveness of the CZM in predicting the adhesive behavior for manufacturing steel/CFRP hybrid parts. The future directions of this study involve further investigation to improve the accuracy of prediction by considering changeable thermomechanical properties for the curing behavior of CFRP and coupled analysis for forming and springback. Additionally, the method suggested in this study will be applied to the design of the manufacturing process of automotive parts consisting of steel and CFRP.

## Figures and Tables

**Figure 1 materials-14-06285-f001:**
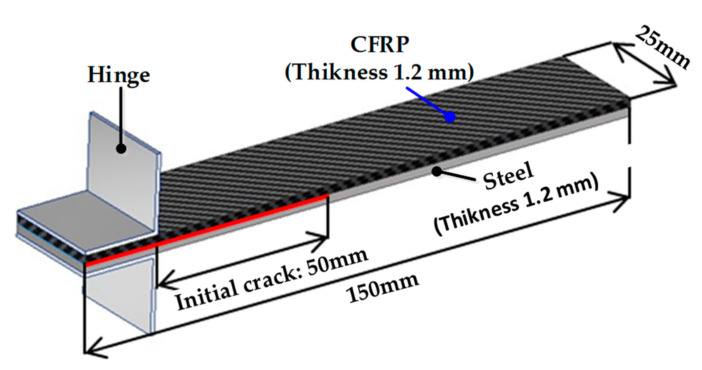
Dimensions of specimen for DCB test.

**Figure 2 materials-14-06285-f002:**
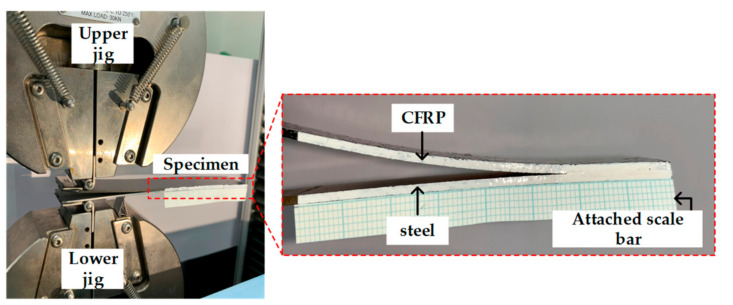
Experimental setup for DCB test.

**Figure 3 materials-14-06285-f003:**
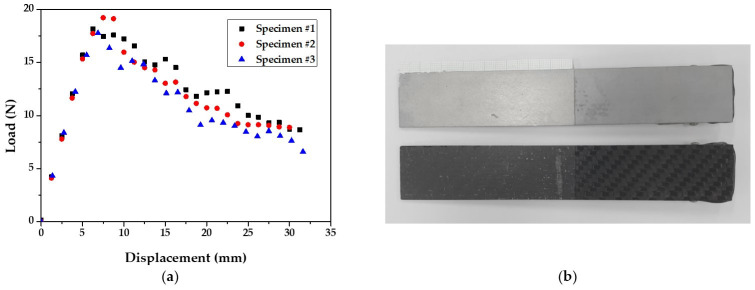
Results of DCB test: (**a**) load–displacement curve; (**b**) failure surface.

**Figure 4 materials-14-06285-f004:**
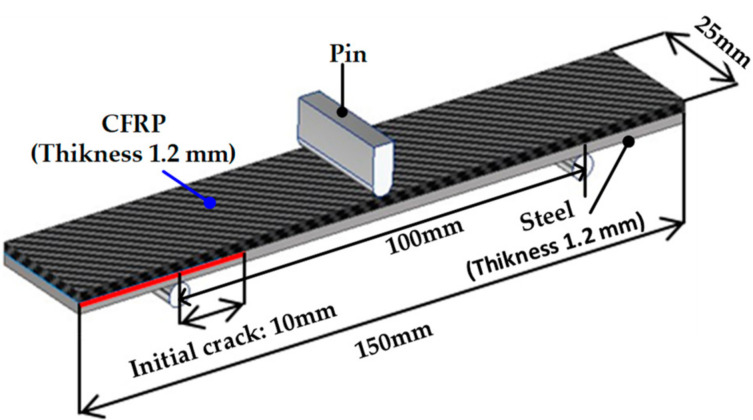
Dimensions of specimen for ENF test.

**Figure 5 materials-14-06285-f005:**
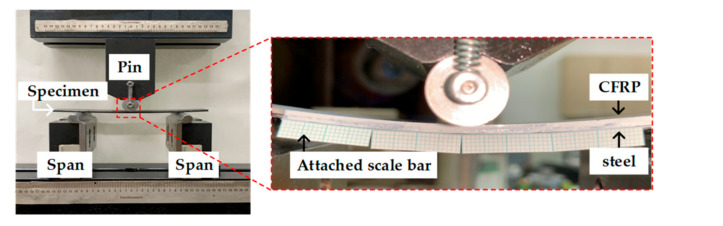
Experimental setup for ENF test.

**Figure 6 materials-14-06285-f006:**
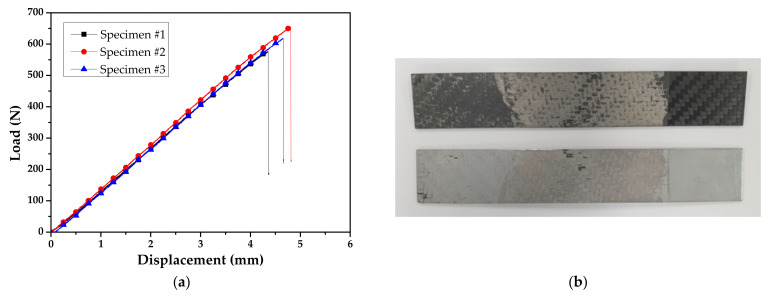
Results of ENF test: (**a**) load–displacement curve; (**b**) failure surface.

**Figure 7 materials-14-06285-f007:**
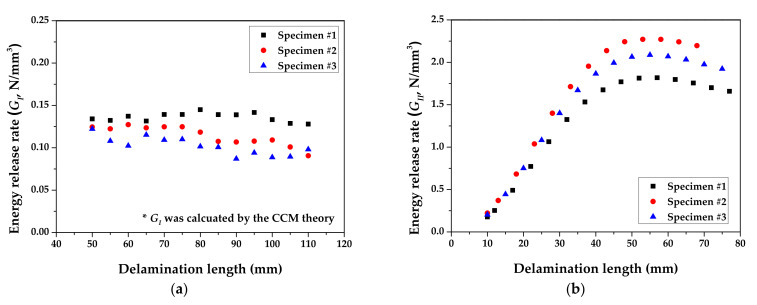
Derived R-curve from experimental results: (**a**) DCB test; (**b**) ENF test.

**Figure 8 materials-14-06285-f008:**
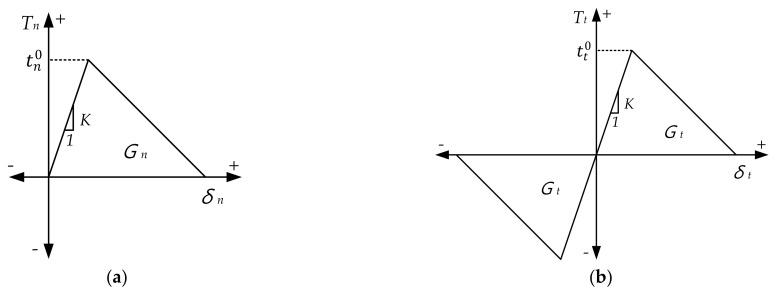
Traction-separation law: (**a**) normal mode; (**b**) shear mode.

**Figure 9 materials-14-06285-f009:**
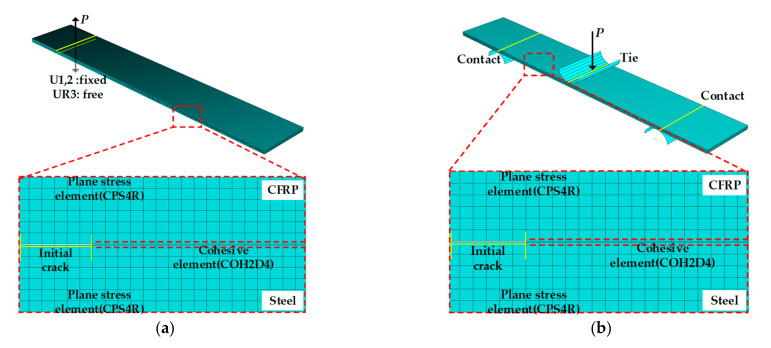
FE model of Steel/CFRP hybrid specimen: (**a**) DCB test; (**b**) ENF test.

**Figure 10 materials-14-06285-f010:**
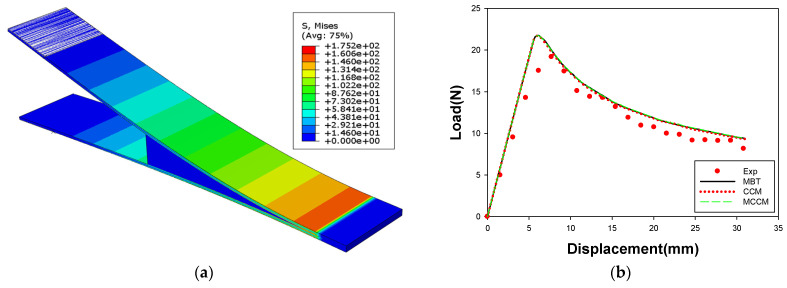
Result of FE simulation for the DCB test: (**a**) distributions for effective stress; (**b**) load-displacement curve.

**Figure 11 materials-14-06285-f011:**
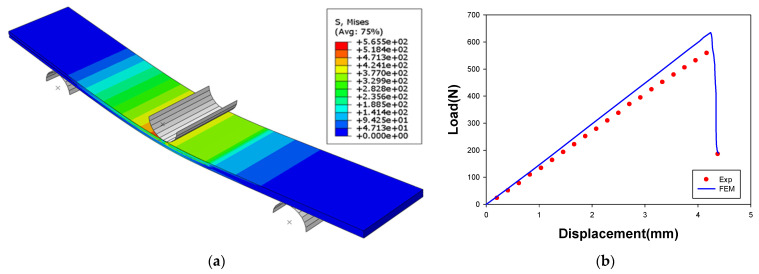
Result of FE simulation for the ENF test: (**a**) distributions for effective stress; (**b**) load–displacement curve.

**Figure 12 materials-14-06285-f012:**
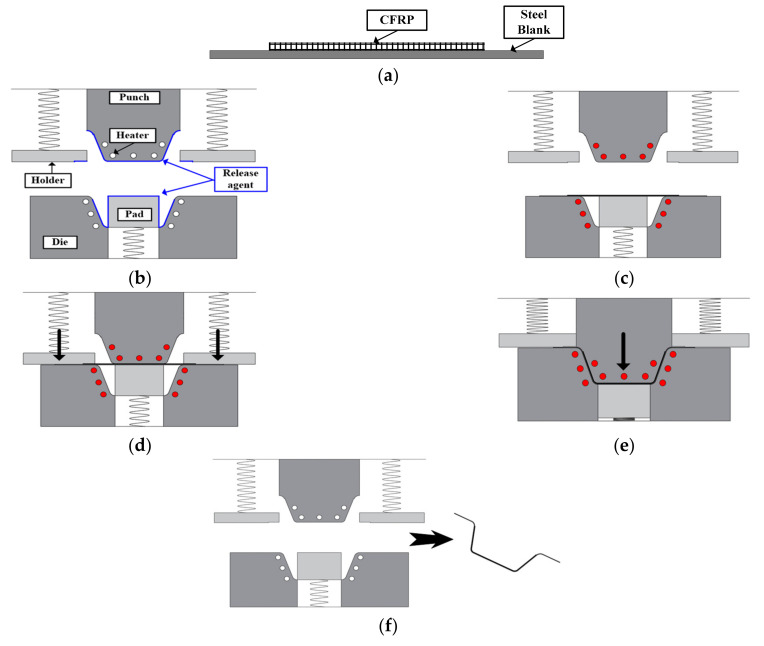
Manufacturing process of Steel/CFRP hybrid part: (**a**) stacking prepreg on steel blank; (**b**) spreading release agent; (**c**) tool heating; (**d**) blank holding; (**e**) forming and curing; (**f**) ejecting.

**Figure 13 materials-14-06285-f013:**
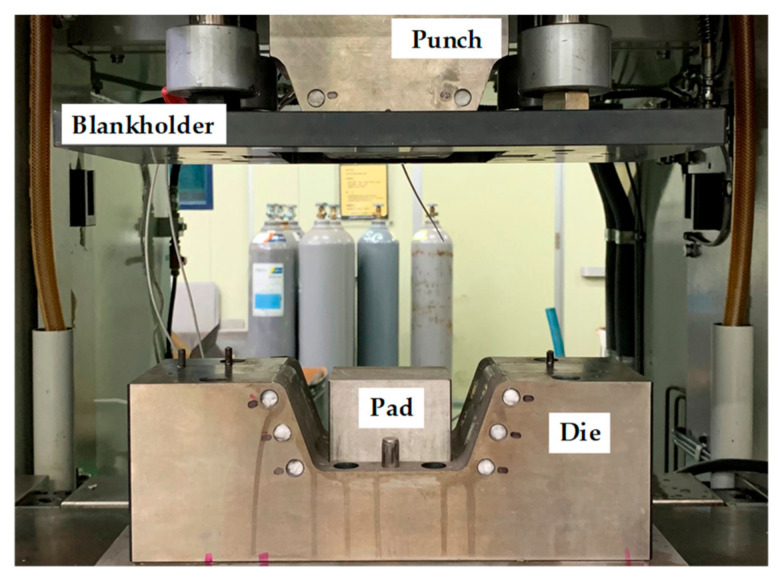
Experimental equipment for manufacturing of Steel/CFRP hybrid part.

**Figure 14 materials-14-06285-f014:**
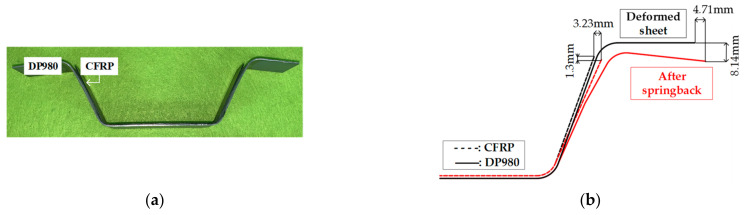
Result of U-shape drawing test: (**a**) manufactured part; (**b**) dimensions of part after springback.

**Figure 15 materials-14-06285-f015:**
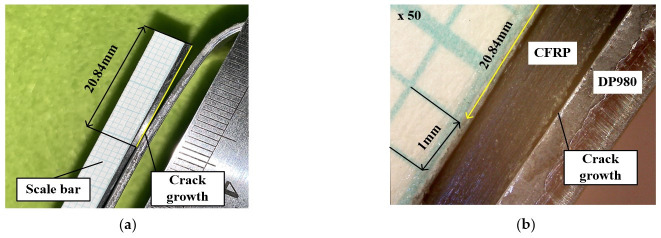
Microscopy images of adhesive failure in hybrid part: (**a**) delamination length; (**b**) crack-tip opening.

**Figure 16 materials-14-06285-f016:**
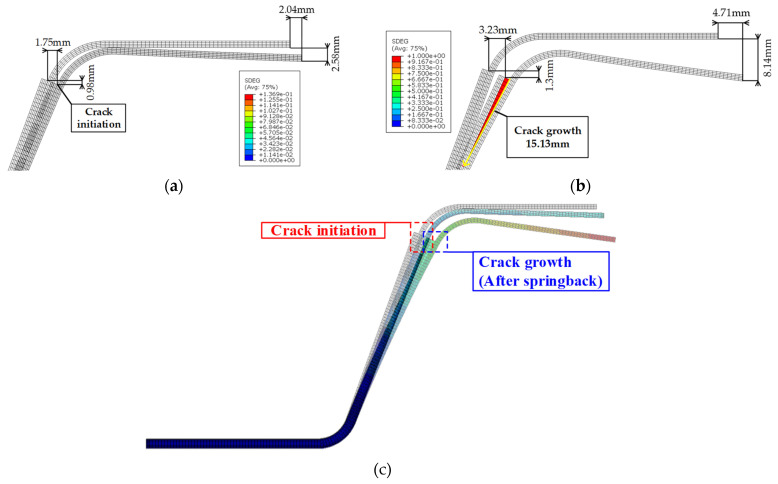
Results of FE simulation to predict adhesive behavior of hybrid part: (**a**) crack initiation; (**b**) crack growth; (**c**) overall shape of deformed part.

**Table 1 materials-14-06285-t001:** *G_I_* and *G_II_* for steel/CFRP part.

Specimen	GI			GII
	MBT	CCM	MCCM	
1	0.144 N/mm3	0.142 N/mm3	0.145 N/mm3	1.658 N/mm3
2	0.127 N/mm3	0.126 N/mm3	0.128 N/mm3	2.196 N/mm3
3	0.119 N/mm3	0.118 N/mm3	0.120 N/mm3	1.972 N/mm3

**Table 2 materials-14-06285-t002:** The material properties for steel and CFRP used in FE simulation [21].

Properties for CFRP	Value
Elastic modulus in fiber direction 0° (GPa)	65.01
Elastic modulus in fiber direction 90° (GPa)	65.01
Elastic modulus in thickness direction (GPa)	3.1
Shear modulus in 1–2 (GPa)	12.69
Shear modulus in 2–3 (GPa)	1.38
Shear modulus in 1–3 (GPa)	1.38
Poisson’s ratio in 1–2	0.13 [15]
**Properties for Resin**	**Value**
Elastic modulus (GPa)	3.1
Tensile and shear strength (MPa)	85
Penalty stiffness (MPa/mm)	155,000
Poisson’s ratio	0.3
**Properties for Steel**	**Value**
Density (ton/mm^3^)	7.8E−8
Elastic modulus (GPa)	190.3
Ultimate tensile strength (GPa)	1.2
Yield strength (MPa)	637.7
Poisson’s ratio	0.33

## Data Availability

The data presented in this study are available on request from the corresponding author and the first author on reasonable request.

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
