# Peer review of "Prediction of the Delamination at the Steel and CFRP Interface of Hybrid Composite Part"

_materials, 2021, doi:10.3390/ma14216285_

Round 1

Reviewer 1 Report

This paper conducts an experimental and numerical investigation on delamination behavior of a composite made of a steel plate and CFRP panel. The paper can be considered for publication after a major revision is made. Following comments are for the authors to take into account in their revision.

  1. Much information in their experiments was missing. Following should be added: what were the specimen thicknesses in the DCB and ENF tests? In the line 69, the thickness of the steel plate and the CFRP panel was 1.2t, but without a specification for the t. Further, what was the span distance in the ENF test (in Fig. 4)? What was the unit for the GI and GII in Table 1 and, possibly, elsewhere? The measured data from specimens 1 and 2, especially for GII, were in relatively large deviations. Explanation for this should be provided.
  2. It is assumed that the CFRP panel was made of unidirectional prepregs in 6 plies (in the line 70). Its lamination detail should be provided. Furthermore, it is better to provide the 5 elastic moduli of a single unidirectional ply.
  3. As pointed out by the authors, CZM has a difficulty in determination of its input data. The authors finally chosen the data, but without a description on how they did, by comparing the predicted with measured DCB curves to determine the data? Please explain.
  4. There were two FE discretization regions on top of the crack initiation in Fig. 15(a), which seemed to be different from the actual structure shown in Fig. 13(a) and 14(a). From Fig. 13(a) and Fig. 14(a), only the metal plate was extended to the top region. The CFRP part was within the U-shape. The authors need to provide more information on their FE simulation.
  5. A similar to CZM method was proposed by Huang et al. recently (Engineering Fracture Mechanics, 238: 107248, 2020), in which no adjusting on the input data is necessary. The authors should apply this method to the analysis of their problem, and should compare the predictions by the CZM and Huang’s method with the experiments.

Reviewer 2 Report

This work presents interesting numerical investigations predicting adhesive behavior in steel and carbon fiber reinforced plastic (CFRP) hybrid parts based on the cohesive zone model (CZM).
Numerical tests on classical DCB and ENF loading conditions were performed. The paper is interesting and deserves to be published in this journal.
Some comments and suggestions were reported.
The figure and the tables are clear.

Even though the gap that you are going to cover is well explained, I suggest reporting the paper's outline at the end of the introduction sections.

The literature review needs to be improved. I recommend to reference paper about CZM, moving mesh methods as well as XFEM:
10.1016/j.compositesb.2016.04.047
10.1016/j.compositesa.2015.10.007
and similar works.

Why is Traction separation low reported only for mode I?

More details about the degradation behavior of the interfaces adopted are required for the sake of clarity.

Why are you using a 3d model instead of a 2d model? Can you please explain this choice? What are the useful outcomes that you get from this modeling assumption?

The conclusion must be enriched with some comments about the numerical simulations adopted and mentioning future development.

Round 2

Reviewer 1 Report

Majority of my previous comments was addressed. But, how the material data used for the CZM simulation (called the penalty stiffness and peak stress in the line 161) were determined was not answered. The authors stated that the data were assumed. Is an arbitrary assumption for those data applicable? If yes, the authors should provide their simulated results obtained from another, arbitrarily assumed pair of data for the penalty stiffness and peak stress; if not, the authors should explain how they attained their “correct assumption”. Furthermore, “the penalty stiffness and peak stress” should be defined using schematic diagram of Fig. 7, with more indications on the figure if necessary, and should be listed in Table 2. In addition, the pure matrix property data such as elastic modulus, Poisson’s ratio, tensile strength, shear strength etc. should be provided in Table 2.     

Reviewer 2 Report

The paper can be accepted in the present form

Author Response

Thank you for you kind comments.